# Combining Gamification and Augmented Reality to Raise Interest in Logistics Careers

Lisa-Maria Putz-Egger [1],* , Denise Beil [1] , Silvia Dopler [1] and Jeremiah Diephuis [2]

1 Department of Logistics, University of Applied Sciences Upper Austria, 4600 Steyr, Austria
2 Research Center Hagenberg, University of Applied Sciences Upper Austria, 4232 Hagenberg, Austria
* Correspondence: lisa-maria.putz-egger@fh-steyr.at

**Abstract:** The logistics and transport industry is currently facing the major challenge of having a global shortage of skilled workers. To address this challenge, this paper evaluates the application of gamification in combination with augmented reality (AR) as a new approach to attract the interest of people of all ages to the logistics sector. The aim of the paper is to determine whether a gamified AR-based application called *Logistify* is a feasible approach to make logistics jobs more attractive. We used a qualitative approach in three phases by collecting and analysing data from different perspectives of players, teachers, instructors, and programmers about the application: (1) analysing game characteristics with programmers and workshops instructors, (2) collecting feedback from players and teachers, and (3) evaluating game scores. The evaluation shows that gamification in combination with augmented reality is a promising tool to attract people to the logistics sector and to change their perception of logistics professions. It can be concluded that the gamified AR approach is capable of increasing interest in jobs in a particular sector.

**Keywords:** augmented reality; gamification; green logistics; logistics careers; lack of logistics personnel





## 1. Introduction

Logistics, including the major function of transport, is a basic requirement for global trade, ultimately representing a critical enabler of worldwide growth and corporate success [1]. The logistics sector plays a vital role in the economy, which contributes 7% to the total GDP and employs more than 11 million people in the EU-27 [2]. Despite automation and digitalization, the vast majority of logistics jobs require the presence of humans. Logistics activities are highly labour-intensive on both operational and managerial levels [3], leading to a high dependency from companies on the availability and qualification of skilled personnel [2,4]. For years, there has been a shortage of qualified personnel in the logistics industry, which has further tightened during COVID-19 times [5]. Moreover, the logistics industry failed to position itself as an attractive sector with interesting career opportunities resulting in a lack of young talent [5,6]. Despite being a growing, innovative, and viable industry, it remains a challenge for the sector to communicate and popularize jobs [2,6].

An approach to get people engaged with logistics skills is gamification, in combination with augmented reality (AR) [7–9]. In general, gamification aims to engage people through game elements such as competitions, feedback functions, or social elements in a non-gaming context [10,11]. Gamification can be efficiently used to motivate people by rewarding a certain behaviour, to influence people's attitude about a certain topic such as sustainability [12,13], to increase educational performance [14,15] and to promote desired behavioural changes [16]. The benefits of adding AR technology to support endeavours based on gamification is two-fold. First, the incredible success of AR-based games such as *Pokémon Go* demonstrate the potentially large-scaled effects such games can have on people [17]. Second, logistics is one of the fields in which AR applications have been

actively employed [18] and therefore it is an effective way for individuals to familiarize themselves with the capabilities of logistics jobs.

Since several studies have already demonstrated that gamification combined with AR is an impactful method to increase interest and engagement in a certain area [19–21], this paper has a different aim. Most of these studies analyse the effectiveness of gamified AR to increase interest, learning motivation, knowledge transfer, and awareness of a particular topic [19,20,22,23]. Yet, there is a gap in using this approach to increase the attractiveness of jobs in a certain sector. Since there is a shortage of skilled workers [6], especially in the logistics sector, there is a need to investigate the effectiveness of gamified AR in this context. Current studies dealing with gamified AR in the logistics context mostly focus on increasing current job satisfaction (e.g., through gamification of a warehouse environment) rather than on attracting new employees [7–9]. Therefore, this paper aims to investigate whether this is also an effective method to promote logistics careers by using a gamified AR app called *Logistify*.

Based on a three-phase methodology adapted from [24], the first phase involves developing the app based on the expertise of programmers and workshop leaders within focus groups. Workshop leaders are experts in the field of logistics who are able to address the current shortage of skilled workers in this area appropriately [25,26]. The developed app is tested while workshop leaders observe the players while using the app. Feedback from these observations is gathered by a questionnaire and analysed using a thematic analysis approach according to [27]. In phase two, the gamified AR app is tested with students and teachers in a workshop environment. A semi-structured survey is used to analyse the effectiveness of *Logistify* in attracting people to logistics jobs. Phase three focuses on analysing data from the game-based app to gain insights into user motivation and performance.

The paper is structured as follows. Section 2 discusses the theoretical background and practical environment of augmented reality, gamification, the combination of both, and the description of our gamified AR app *Logistify*. Section 3 gives an overview of the methodology by describing our three phased approach. Section 4 outlines the results based on the three phases. Finally, Section 5 discusses the results, and finishes with a conclusion and an outlook on future research needs.

## 2. Theoretical Background and Practical Setting

### 2.1. Augmented Reality

Augmented Reality (AR) is traditionally seen as part of the Reality–Virtuality Continuum [28] and typically refers to applications that superimpose virtual content over a real-world camera feed, thus providing a hybrid experience. To integrate this virtual content into the real-world, printed visual markers are frequently used to serve as anchors for digital content, allowing it to be viewed from any angle, appearing as part of the real world. Recent smart devices such as smartphones, tablets, and mixed-reality headsets also often incorporate technologies such as depth cameras and motion-tracking sensors to enable this integration without the use of such visual markers [28,29].

AR technology still involves several limitations that need to be considered. AR requires the use of a camera and integrated image processing technologies and is heavily dependent on the quality of the lighting and surrounding environment, just as in the case of any camera. Even when a printed marker is used to serve as a stable anchor for virtual content, reflections and poor lighting can result in poor performance and an unsatisfactory gaming experience. The number of markers that can be utilized simultaneously is limited to the processing power of the device, typically being constrained to five or six simultaneous markers. In addition, any kind of occlusion, such as a hand passing in front of the devices' camera, can briefly disrupt the tracking and potentially interfere with the augmented experience. In addition, most AR technologies simply superimpose virtual objects over the real-world vision, resulting in objects simply appearing over such an occlusion, thereby destroying the illusion of a hybrid world [30].

As smart devices are becoming more technologically advanced, actual uses of augmented reality beyond the fields of applied research are getting more common, particularly in the fields of mobile gaming, art, education, industrial applications, and logistics [29,31]. Although educational contexts have primarily utilized digital and/or animated extensions of examples in printed textbooks [32], AR applications can now be found as standalone apps that can be used indoors, outdoors, on table tops, or basically anywhere a smart device with a camera can be used [33]. AR technology has a fairly unique, fascinating quality that can be effectively employed for playful learning activities, allowing users to interact with a tiny virtual world right in front of them [34].

### 2.2. Gamification

Gamification in general is used, among other things, for educational purposes as well as to influence people's attitudes and behaviour [35,36]. Various studies have demonstrated the positive results of well-designed gamification on learning performance [36–39]. Moreover, previous results suggest that gamification influences the subjective personal attitude towards a topic and minimizes the barrier of thinking about new career paths [36]. To name some examples, Pérez-Manzano and Almela-Baeza [40] used gamification-based applications to raise the interest in science and to promote scientific careers, Ansted [41] proposed to use gamification in a broad field of application for career guidance, and McGuire et al. [42] created gamified workplace simulations to enhance students' motivation and awareness of career opportunities. According to Bhalerao et al. [43], gamified career decision-making systems can turn the career selection into an engaging process. Their review also found that there is a lack of research on the use of gamification in career choice. Putz et al. [26] conducted a pre-test post-test survey and found that gamification is suitable to increase the attractiveness of green logistics jobs. One of the most common techniques in gamification is the utilization of points and a leaderboard that ranks the player performance. Such scoring systems not only serve to motivate players to perform at higher levels, but also encourage them to play the game again and provide players with a quantitative assessment of their progress [44]. This data can be used as a reference to assess player motivation and learning performance, at least to some degree [45].

### 2.3. Gamification and AR Combined

Gamification in combination with AR is used in the logistics industry to support workers, e.g., for order picking [46] as well as for educational goals [47]. Plakas et al. [48] agreed that the use of AR and gamification in the picking process led to an increase in efficiency and job satisfaction and subsequently the overall performance. Noreikis et al. [49] found that a well-designed combination of AR and gamification encourages active engagement and provides a higher learning achievement based on the quiz results achieved. In addition, players achieved a higher level of enjoyment and social experience [49]. Savela et al. [50] found contradicting results in an experiment with and without AR, demonstrating that the participants using AR and gamification recognized the entertainment value and learning opportunity but did not achieve an increase in learning performance compared to the group without AR. Even if many AR users reported a high level of engagement, ambition, as well as a lower level of tiredness, AR users scored significantly lower compared to non-AR users [50].

### 2.4. Description and Use of "Logistify"

The augmented reality application *Logistify* aims to attract people to the logistics sector and educate them about green logistics. In addition to comprehensive background information on the various modes of transport and the representation of job profiles in the logistics sector, a high priority was given towards transferring information about the modal shift to green transport modes. *Logistify* is divided into three games (1) "Choose the Transport", (2) "Transport Chains", and (3) "Professions". Figure 1 shows the entry screen for all three games. For the first game, the AR marker is optional, meaning the game

can be performed without AR, while for the second game augmented reality markers are obligatory for the game flow to indicate a transport choice as correct or incorrect. In the third game, no AR application is integrated. *Logistify* works with any tablet or smartphone running on iOS 12.0 or later, as well as Android 10 or later and utilizes printed materials as visual AR markers for Game 1 and 2 as shown in Figures 2a and 3a. The respective AR views for the games are shown in Figures 2b and 3b.

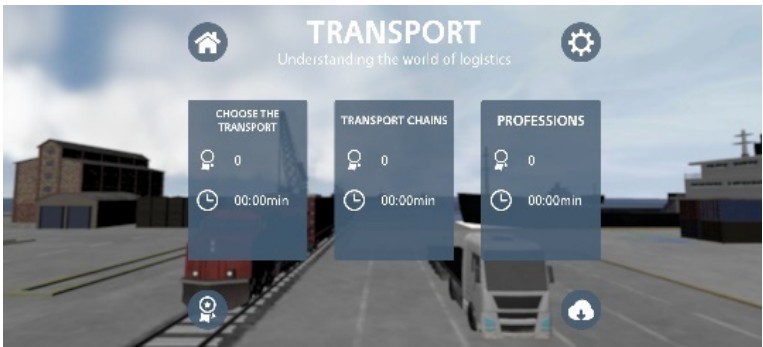

**Figure 1.** Main menu with overview of the games (Logistikum).

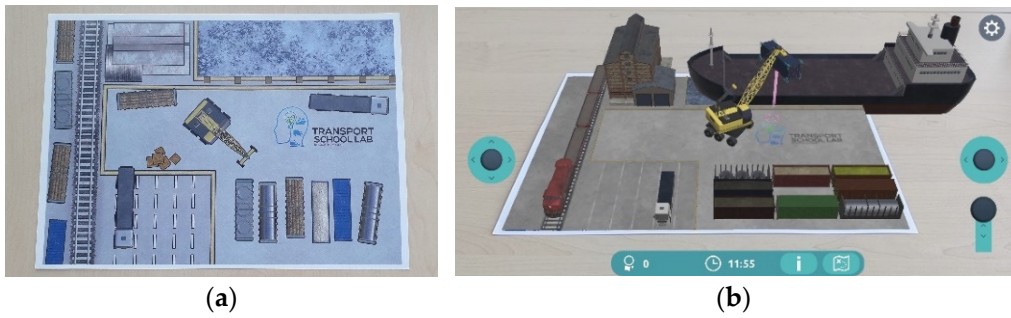

|(**a**)|(**b**)|

**Figure 2.** AR marker of Game 1 "Choose the Transport": (**a**) game map in real world; (**b**) game map visualized on the tablet with augmented reality (Logistikum).

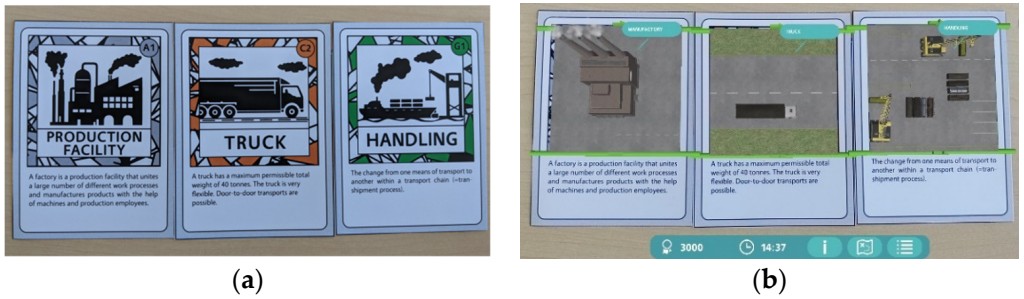

|(**a**)|(**b**)|

**Figure 3.** AR markers of Game 2 "Transport Chains": (**a**) example game cards in the real world; (**b**) example game cards viewed on the tablet with augmented reality (Logistikum).

In Game 1 "Choose the Transport" (Figure 2), players must operate a crane and transfer containers with different types of goods to the most appropriate mode of transport based on the type of goods, the quantity to be transported, the environmental footprint, and the distance required. By lifting a given container with the crane, the player gets an overview of relevant transport details (such as loading and unloading location, desired transit time, type of goods) and can estimate the geographical distance on a map. In addition, background information is provided about the environmental friendliness of the three modes of hinterland transport (inland vessel, train, truck). The goal is to efficiently select the best transport route based on different goods, with special attention to the appropriate means of transport (truck, train, or barge). In the process of the game, it is important to

find out which goods can be transported by sustainable modes of transport, such as inland waterways or rail, in order to reduce the need for road transport.

For game 2, transport chain cards need to be sorted to build a transport chain based on the instructions. The game deals with the planning of multimodal transport chains (pre-carriage, main carriage, on-carriage) from the raw product to the end customer. The participants can choose from a total of five different transport chains, based on different sectors (e.g., automobile industry, steel industry, fashion industry). After scanning the cards with the camera, it detects whether the order of the cards is correct, animates the scene, and marks the cards with green (correct) or red (wrong). The focus is on the optimal utilization of the available resources (which are available in the form of cards) and the consideration of the handling phases. Figure 3 part demonstrates an example of correct results for the transport chain in game 2.

The third game "Professions" serves to familiarize the users with logistics jobs. This game is designed in the style of a message chat which the players utilize to find out which tasks are performed by different logistics personnel, ranging from operational, such as operating logisticians or forwarding merchants to managerial levels such as logistics engineers or logistics managers. The goal of this game is to assign the appropriate tasks and later also characteristics to the selected professions. At the end of the game, the participants receive a list of possibilities to perform the selected profession.

## 3. Method

This research follows a multi-faceted stakeholder approach to identify learnings from the use of an augmented-reality gamified application. In this paper, we used a qualitative method following Ihamäki and Heljakka [24]. Figure 4 summarizes the methodical approach of this paper. We collected and evaluated data from several perspectives about *Logistify* in three phases: (1) developing, refining, and analysing game characteristics with focus groups and a semi-structured questionnaire, (2) collecting semi-structured feedback from players and teachers using an online questionnaire, and (3) evaluating game scores achieved by the players in *Logistify*.

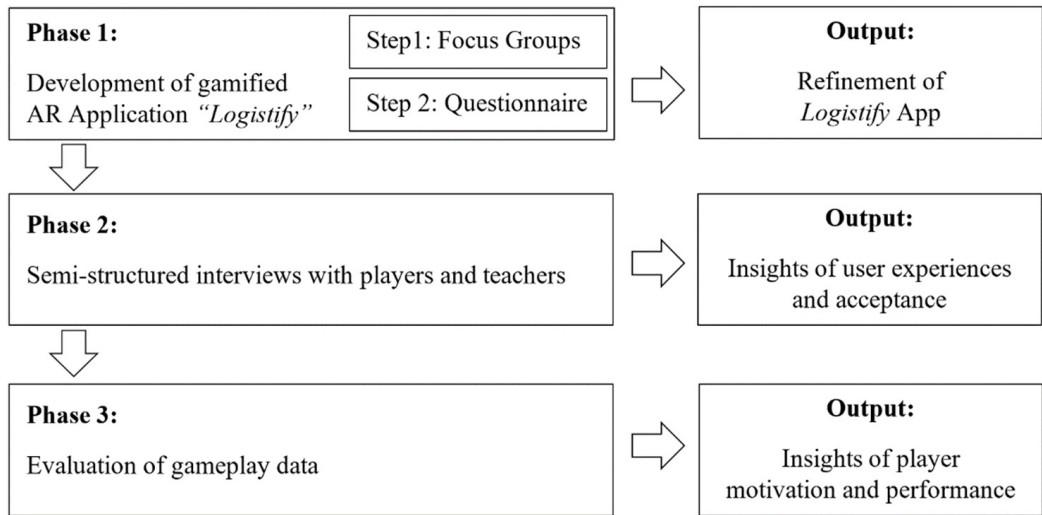

**Figure 4.** Methodical approach modified by Ihamäki and Heljakka [24].

The remainder of this chapter describes the details regarding each phase. Concerning ethics procedures, we followed the regulation (EU) 2016/679 of the European Parliament and of the Council of 27 April 2016 [51] as well as the rules of good scientific practice, such as the European Code of Conduct for Research Integrity [52], the OECD's Best Practices for Ensuring Scientific Integrity [53], and the European RESPECT Code of Practice [54].

The first phase served to develop, refine, and analyse the application based on two steps. Step 1 comprised the realization of three focus groups with programmers and work-

shop instructors. The workshop instructors have professional backgrounds in logistics, research, and pedagogy and regularly conduct one-day workshops including *Logistify* as a major part of the workshop. Two programmers and four instructors were invited for each of the three focus groups. These focus groups clarified how the app should be developed in terms of design, quality, and, most importantly, verification of feasibility in an interdisciplinary environment. While the instructors provided the expertise regarding logistics careers, the programmers assessed the technical feasibility. To facilitate the availability of *Logistify* on multiple mobile platforms, the games were developed using the *Unity* game engine, which provides comprehensive support for multiple platforms. Currently, the app is available on both *iOS* and *Android* system in the *Apple App Store* and *Google Play Store*. The AR framework Vuforia was used to implement marker-based tracking. In theory, the games would also support *ARKit* and *ARCore* frameworks for the tracking components, which would likely provide better performance; however, since older smart devices do not always support these frameworks, *Vuforia* is still used. The second game, "Transport Chains", also features a custom modular tracking extension, which allows multiple cards to be assembled in any order, and through AR the accuracy of each card is verified.

Step 2 involved the development and analysis of a questionnaire targeted at instructors to document their experiences with players by observing and mentoring them in the workshops. Data from three workshops was collected and was used to uncover potential gaps and areas for improvement, which led to further revisions and adjustments to the app. The questions can be found in Appendix A. The questionnaire consists of four open-ended questions. The answers have been evaluated using inductive thematic analysis [27]. The statements of the workshop leaders were paraphrased and then semantically grouped. From this, superordinate thematic groups are derived and patterns are identified.

Phase two comprised collecting feedback from players and teachers after using the *Logistify* games in a workshop environment. The app was tested in one-day workshops within the framework of school and adult education programs. The aim of these workshops is to bring people closer to logistics, highlight the importance of logistics careers, and raise awareness in the field of sustainable transport alternatives. For the use of the *Logistify* app, the workshop participants were divided into groups of two, which creates an interactive setting and promotes communication as well as social interaction between the players. After a short introduction and explanation about the use of the required materials, the tablets were distributed amongst the workshop attendees. About one hour was needed in total for the briefing, the playing time, and a discussion of the results. Each of the three *Logistify* sub-games took about 15 min to be completed by the participants. After the use of *Logistify,* the workshop participants were encouraged to complete a short semi-structured feedback form after playing each of the three *Logistify* games. The questions were (1) "How did you enjoy *Logistify*?", (2) "Which highlights did you experience during *Logistify*?", and (3) "Which opportunities for improvement do you see for *Logistify*?" For the first question, a five-point Likert scale was used, with the number 5 indicating the player "did not enjoy the game at all" and the number 1 indicating that the player "fully enjoyed" the game. The other two questions had an open format to be answered with text in a comment field. The teachers were asked the following open questions: "Which insights did you gain from the supervision and observation of the players during the use of *Logistify*?". In total, 39 players and four teachers participated in the survey.

Phase 3 included the evaluation of gameplay data such as average playtime and individual game scores that were achieved by the players. Basic gameplay data was collected in the AR game using the *Unity3D* game engine, saved and visualized using Microsoft's *PlayFab* platform. Post-workshop game data analysis was performed using the aggregate gameplay data. Personal data such as names were not collected for this analysis as an effort to ensure the anonymity of all users.

## 4. Results

The following sections describe and discuss the results from each of the three methodological phases. By the means of the conducted thematic analysis (phase 1), we identified major differences depending on the age of the participants of the survey. Hence, throughout the whole chapter, we will examine and highlight these differences. To clarify, younger users, so-called digital natives, who are born into the digital age, had little to no difficulty using the app. Older users, so-called digital immigrants, who acquired the use of digital technologies at some stage during their adult life, faced several challenges in using the app [55].

### 4.1. Phase 1: Workshop Instructors and Programmers

The feedback from the three focus groups (step 1) with the programmers and workshop instructors led to some minor adjustments at the beginning regarding content and display of information. The major progress in the development of *Logistify*, focusing on the use of AR elements, was made by analysing the observations of the workshop instructors (step 2) to identify potential gaps and areas for improvement. The main insights from these questionnaires are mapped in Figures 5 and 6 and described for each question in detail below.

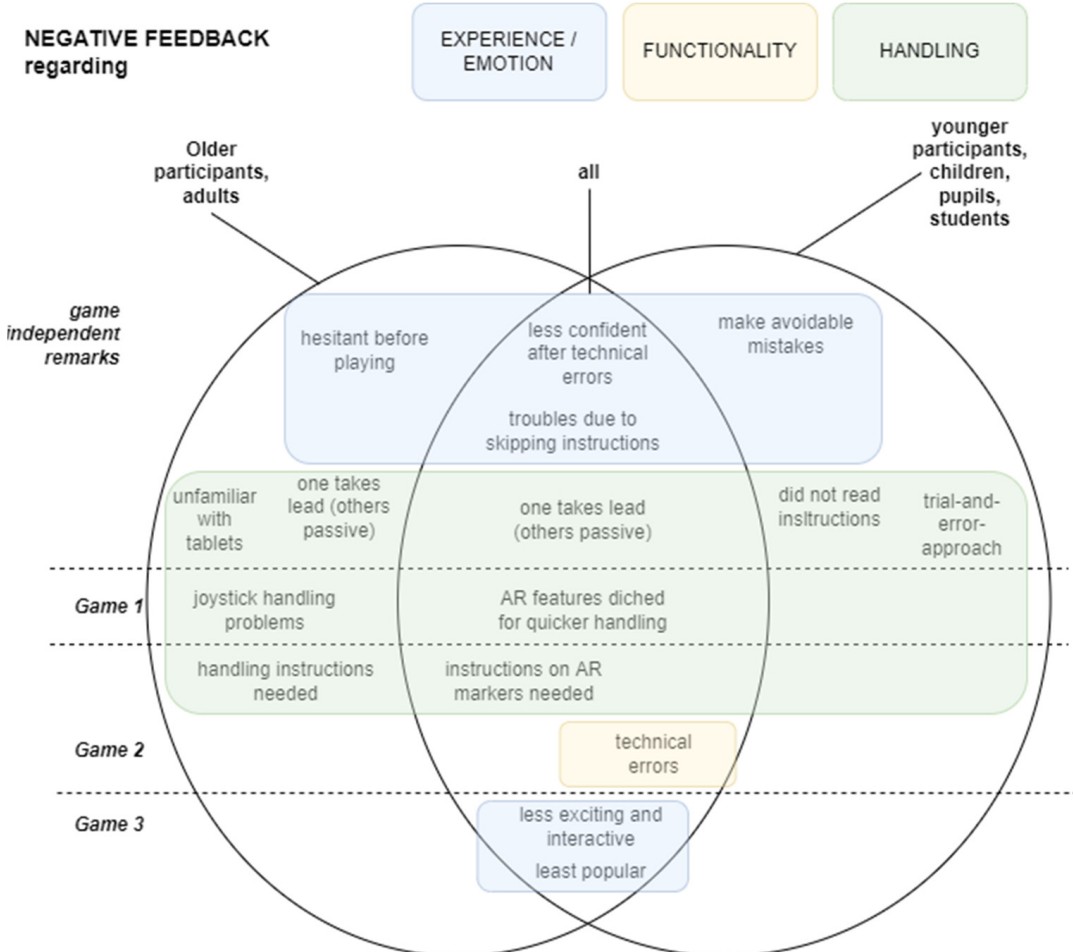

**Figure 5.** Results of the thematic analysis regarding potential improvements for *Logistify* (Own illustration).

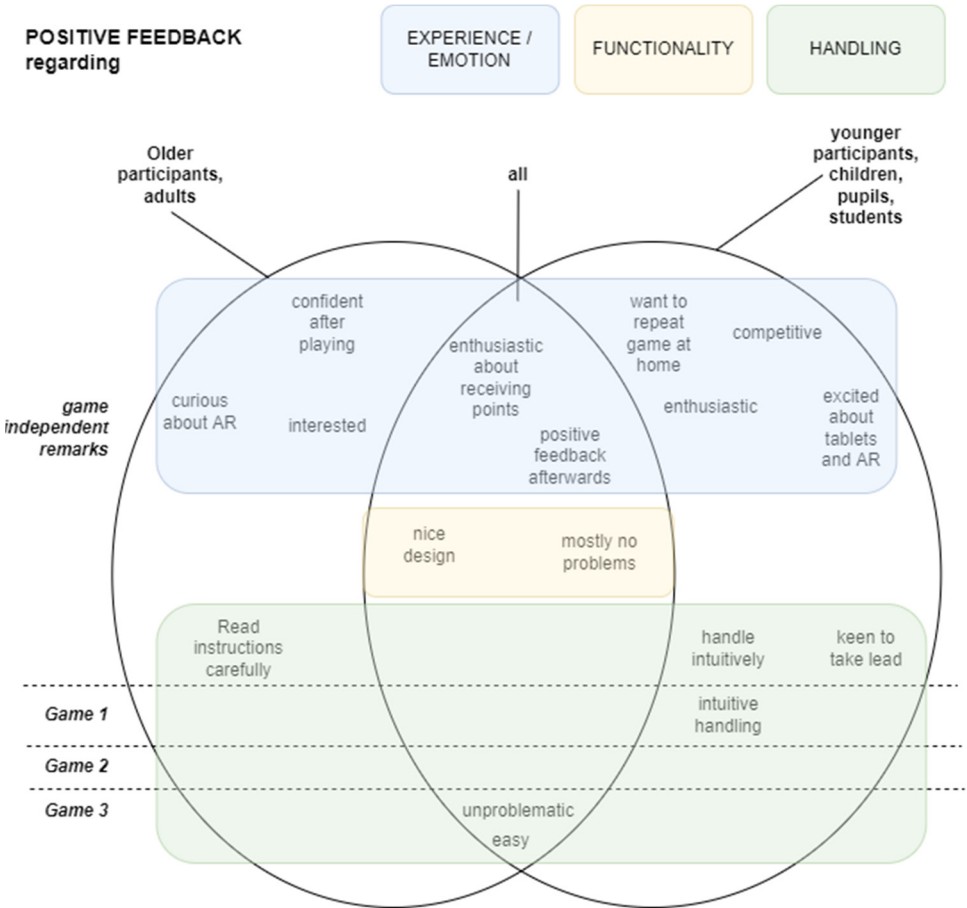

**Figure 6.** Results of the thematic analysis regarding positively annotated observations (Own illustration).

4.1.1. How Did the Players Cope with Game 1 "Choose the Transport", Game 2 "Transport Chains", and Game 3 "Professions"?

In general, most players did not experience major challenges with the use of *Logistify*. Since the workshops included some instructions on how to proceed, the users were quickly able to familiarize themselves with the overall gameplay. Digital immigrants sometimes had difficulty operating the virtual joysticks in game 1, which is needed to operate the crane and the containers, whereas the digital natives rarely had any difficulties. The younger the players were, the more they tended to skip over the instructions without reading them. Instead, the players just clicked through the instructions, which led to some uncertainty about the actual procedure to be followed. Game 2 often required some additional explanation since players did not immediately understand how to arrange the cards. At first, they scanned a card and then put it away instead of trying to build a complete chain. This aspect was tackled in a revised version of *Logistify* with an in-game instruction and hints. Game 3 was unproblematic for users of all age groups and genders.

The overall insight from this question for the development of *Logistify* and the workshop was to provide digital immigrants with more time to familiarize themselves with *Logistify* and to point out that reading the instructions is necessary to follow the correct process of *Logistify*. Digital immigrants typically invest more time to familiarize themselves with the handling of *Logistify* than digital natives. In addition, instructors observed that digital natives exhibit more competitive behaviour than digital immigrants. To elaborate, the competition among younger users is higher than among older users, for example, regarding score achievement. After the games, the scores were often compared, which encouraged the younger users to play further rounds of the app.

### 4.1.2. How Did the Players React to the First Use of the Augmented Reality App Characteristics? (Before and After)

The group of digital natives were excited to be allowed to play with tablets and use *Logistify*. The fact that AR was involved did not appear to affect the overall excitement level, but the appeal of playing a game was evident. Digital natives were evidently enthusiastic before and after playing *Logistify*. Digital immigrants more often hesitated before starting to play since they seemed worried about not being able to effectively control the gamified-AR application. Indeed, after playing *Logistify*, most of the players expressed their enjoyment of playing it. Several users noted that the realistic representation of the terminal in Game 1 made the handling process very tangible and allowed them to better envision how the handling and planning of transports actually occurs.

Players were less positive when it comes to technical errors that hampered their gaming experience. The most common difficulties were due to technical handling errors experienced by the users in game 2, which required players to assemble cards into a transport chain. It was observed that technical errors, which sometimes appear due to sensitivity of the cameras to lighting (as mentioned in Section 2), immediately disrupted the attention of the players. For the first game, the use of the AR map is optional, meaning that the game works with and without AR function. In particular, highly motivated groups of mainly digital natives decided not to use the AR map, since they assumed this would save time and they would therefore receive some extra points by only using the traditional 2D view.

### 4.1.3. Which Insights Did You Gain from the Supervision and Observation of the Players during the Use of Logistify?

Most of the time, players needed no help, while digital immigrants often needed to see how *Logistify* works first and then were willing to try it themselves. Digital natives sometimes were keener to play and argued about who gets to play first. Most players did not read the instructions. Therefore, it is important that *Logistify* has an intuitive user interface. In general, digital immigrants read information or texts more carefully than digital natives did. Moreover, the instructors assumed that digital immigrants did not want to miss anything and were afraid of making mistakes, whereas digital natives more frequently followed the principle of "trial and error". Throughout the gaming sections during the workshops, the instructors noticed that interest in the handling of containers, decision making in the choice of transport modes, and the establishment of transport chains increased.

### 4.1.4. What Did the Players Like Most/Least?

Players were enthusiastic about receiving points and comparing their scores with the other teams following competition and social interaction, which were important gamification elements. The AR elements in the first and second game were perceived as fascinating, as each scene on each card comes to life featuring an animated step in the transport process. This helped the participants in better understanding the logistical processes in the real world.

Although the design of game 3 is the most familiar, resembling a chat application, it was the least popular game of the three games. In fact, it is considered as a game leading to monotonous tasks, which has to be improved in the next version of *Logistify*. Compared to the first two games, it was evaluated as less exciting and interactive, as the players simply have to drag and drop the answers. In the feedback rounds after each workshop, the players mentioned that game 3 was by far the least engaging.

It must be emphasized that technical errors, which were more likely to happen during AR applications (game 2 in particular) frustrated the players and interrupted their learning experience. Groups who aimed at achieving a high score and faced technical problems were very disappointed.

To summarize, the first phase allowed refinement of the app based on the input given to the programmers by the workshop instructors. Testing the app in three workshops enabled the identification of potential gaps, such as some technical issues, the description of the instructions, and some design features. After this testing phase, the next phase will continue with the analysis of personal feedback from students and teachers.

*4.2. Phase 2: Players and Teachers*

Phase two dealt with the analysis of the semi-structured questionnaire presented to students and teachers to find out how the app is evaluated and its benefits for increasing the attractiveness of the logistics sector. The evaluation of the question "How much did you like *Logistify*?" showed that all three games were overall rated very positively by the 39 survey participants. Game 1 "Choose the Transport" received the best score ($\mu = 1.38$, $\sigma = 0.67$), which is a significant positive result. The low standard deviation showed that most participants agreed with giving a good score. Additionally, when analysing the remaining questions of the survey, it became clear that game 1 was the favourite for most participants. Game 2 "Transportation Chains" received the second-best score ($\mu = 1.69$, $\sigma = 1.13$), which also represents a favourable result. Since the standard deviation is quite high, the participants did not fully agree on the rating. Game 3 "Professions" received the lowest score compared to the other games ($\mu = 1.79$, $\sigma = 0.99$), which is still a favourable result. In summary, these results, which are the only quantitative deliverable of the semi-structured evaluation, show that the approach is highly embraced.

The results of question two "Which highlights did you experience during *Logistify*?" is structured based on the three games. In game 1, the playful learning, the AR features and the good game concept were positively emphasized. The participants found that they developed a greater understanding of logistical processes and the importance of green logistics through the additional information while playing *Logistify* than without a gamified environment. Mixed feedback was given on the level of difficulty and usability. This can be attributed to the fact that digital natives were more familiar with *Logistify* than digital immigrants. In game 2, the gamified concept, the augmented reality graphics, and the visualization of the logistical steps on the transport route were positively emphasized. The level of difficulty of *Logistify* was rated differently. Several players stated that game 2 needs a more detailed explanation. This feedback was picked up in an updated version of *Logistify*.

The final question, "Which opportunities for improvement do you see for *Logistify*?", addressed potential gaps. The feedback shows that the players' prior knowledge in the field of logistics chains and transport processes is relevant. If the prior knowledge is limited, the users need to receive some information on transport chains before playing *Logistify*. The third game was rated by the players to be informative and a good source of information about the requirements of the specific logistics careers. However, game 3 was rated as too protracted and more boring than the other two games. Some aspects that could be improved were mentioned, e.g., that the presentation of the advantages and disadvantages of the individual means of transport could be improved or that the solution should be given at the end of each game. These are important hints that will be taken into account in the further refinement of the app.

The results of the questions addressed to the teachers showed that the innovative teaching approach and AR technology were considered as being very contemporary. For most of them, it was the first-time using gamified AR. They argued that the topic of logistics professions and sustainability fit well with the curriculum of their study programs. Games 1, 2, and 3 were rated as very well designed, although handling Game 1 was sometimes difficult, referring to the challenges of operating the crane. The teachers recommended designing pre- and post-preparation material to discuss and consolidate the learned content. Moreover, the teachers stressed that they were not aware of the broad and interesting spectrum of logistics careers.

*4.3. Phase 3: Game Scores*

The mobile application comprises the three different games, and they are meant to be logically separated but still playable in combination with each other. Due to this architecture, the assessment of the three games was different. Each game featured a similar scoring structure, adding points for correct choices and subtracting points for incorrect choices, which were then computed into a total score. Since every single game was structured differently, direct inter-game comparisons are not possible. Game 1 was designed for repeatability with features that are basically the same content albeit randomized for every round. Game 2 featured multiple levels called "transport chains", which allowed players to choose varying levels of difficulty, without requiring them to play all of them or play the chains in a particular order. Game 3 requires the players to click through all introduced careers.

The post-evaluation analysis of player motivation and performance based on *Logistify* game scores proved to be challenging. The intention of the score analysis was to examine whether there is a noticeable similarity between team game performance and general satisfaction with the application. During post-evaluation analysis, we recognized that data in the application was being collected incorrectly and could not be used for this purpose. Since the workshop instructors also documented the results of the participants by hand, this data is used for the gameplay analysis. For game 1, the maximum score is 20,000, and the scores achieved by the participants were generally between 16,000 and 20,000 points. This may be explained by the fact that digital natives played the games repetitively, and thus often achieved the total score. The average score for game 2 is between 6500 and 7500 with a maximum of 10,000 points. For game 3, no scores were documented since the gaming environment is just designed as a message chat. Nevertheless, the process has given insights into how the application needs to be reworked to include such features, and future work will include the restructuring of the analytics functionality.

## 5. Concluding Discussion and Future Research

In this paper, we investigated the impact of a gamified AR app called *Logistify* on peoples' attitude toward logistics professions. In three methodical phases, the approach was refined based on user feedback. Using a semi-structured survey among 39 participants, the general acceptance and learning effect was analysed. Our paper contributes to the limited theory of gamification, AR, knowledge, and career choices using a qualitative approach as a starting point for deeper observations. From a practical point, the paper can be used for teachers, trainers, or programmers as an input for the design of gamified AR applications. The evaluation of the results indicates that the approach of *Logistify*, i.e., to utilize augmented reality in combination with gamification elements, is suitable to engage people with the exploration of logistics in terms of job opportunities and knowledge. Nevertheless, the results show that there are major differences between digital natives and digital immigrants, with the latter experiencing some difficulties in using the app. Based on these results, the workshop procedure and the gamified AR app have the potential to be reshaped to meet the needs of all participants and increase the level of knowledge retention.

Regarding the professions in logistics, both users and teachers were hardly aware of the large variety of professions in logistics. Through the paper, it can be stated that knowledge retention about logistics careers through gamified AR is a successful approach. A lot of participants raised additional questions about how these logistical processes work in companies, how similar they are to the procedures in the terminals or about available career options in logistics. It is an important output that the players liked what they have learned about logistics professions in an interactive way. Similarly, as Ponis et al. [7] and Noreikis et al. [49] found out, it became evident through this paper that knowledge transfer does not feel like "learning".

Additionally, the players agreed that *Logistify* represents an innovative learning approach to transfer practical knowledge about the importance of green logistics and that after playing *Logistify* their interest in potentially following a logistics career path increased.

Figure 7 creates an overview of the conclusion of the usage of the *Logistify* app with the aim of fostering knowledge in green logistics careers.

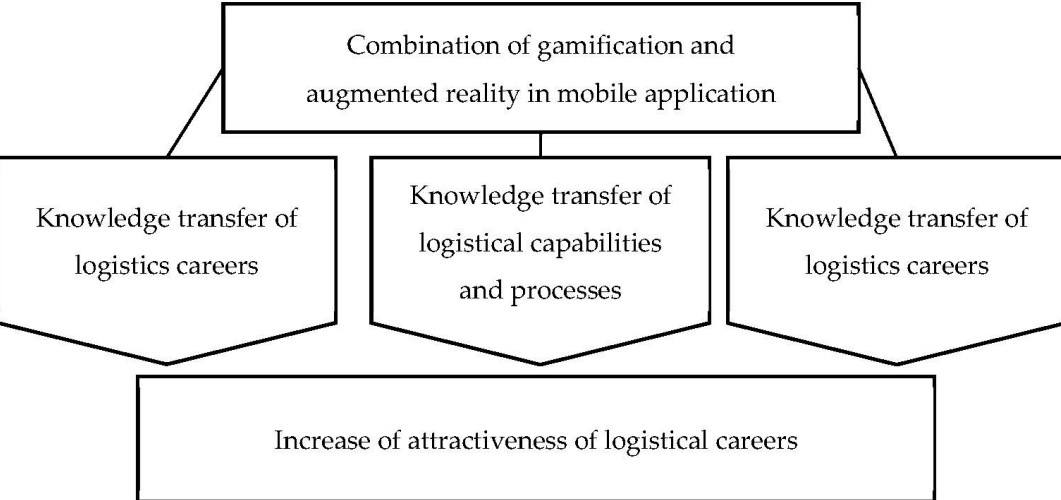

**Figure 7.** Conclusion for the use of gamification and AR to attract green logistics careers (Own illustration).

At the same time, feedback from players during the evaluation process indicates several specific improvements that can be made to optimize the effectiveness of the application. These include better handling of game-related errors, either due to technological limitations or lack of clarity from the user's perspective. Although AR functionality can fail or perform poorly at times, most errors appeared due to an unintended reuse of the printed markers. Ideally, future versions of *Logistify* need to recognize such a use and alert the player to what is happening. In general, improvements to the user interface must be implemented to reduce perceived system errors and to reduce AR-related errors, as these frustrate the players and significantly interrupt the learning process. Moreover, an updated version of *Logistify* needs a re-design of game 3 as well to include a scoring model that can be used for data evaluation.

In terms of game design, we found that digital immigrants tended to be overwhelmed with the general controls of the application and need more time to familiarize with the technical setting. Digital natives did not require explanation about how to use *Logistify* at all.

Future research should include an experiment about the enjoyment and learning performance comparing a setting with and without the AR function. Finally, the self-assessment should be replaced by a measurement following the approach of Savela et al. [50] for learning performance. For example, a quiz could be included to draw inferences based on data about knowledge retention. Since we only used a semi-structured approach to analyse user experiences, it might also be interesting to evaluate the acceptance of the gamified AR approach more quantitatively, especially regarding the various experiences based on age.

**Author Contributions:** Conceptualization, L.-M.P.-E., S.D. and D.B.; methodology, L.-M.P.-E. and D.B.; software, J.D.; investigation, D.B. and J.D.; data curation, S.D. and D.B.; writing—original draft preparation, L.-M.P.-E. and D.B.; writing—review and editing, S.D., L.-M.P.-E. and D.B.; project administration, L.-M.P.-E.; funding acquisition, L.-M.P.-E. All authors have read and agreed to the published version of the manuscript.

**Funding:** This research was funded by the research cooperation REWWay which is funded by viadonau.

**Acknowledgments:** Open Access Funding by the University of Applied Sciences Upper Austria.

**Conflicts of Interest:** The authors declare no conflict of interest.

**Appendix A**

Phase I:

We used the following questions for the feedback of the workshop instructors:

(1) How did the players cope with game 1 "Choose the Transport", game 2 "Transport Chains", and game 3 "Professions"?

(2) How did the players react to the first use of the augmented reality app characteristics? (Before and after)

(3) Which insights did you gain from the supervision and observation of the players during the use of *Logistify*?

(4) What did you observe: what about *Logistify* did the players like most/least?

The programmers observed the use of *Logistify* and were asked similar questions to the ones above.

Phase II:

We used the following questions for the feedback of the players:

(1) How did you enjoy *Logistify*? (5-point Likert-scale)

(2) Which highlights did you experience during *Logistify*?

(3) Which opportunities for improvement do you see for *Logistify*?

We used the following questions for the feedback of the teachers:

(1) Which insights did you gain from the supervision and observation of the players during the use of *Logistify*?

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
