# Peer review of "Combining Gamification and Augmented Reality to Raise Interest in Logistics Careers"

_applsci, doi:10.3390/app12189066_

Round 1

Reviewer 1 Report

My concern is about the methodology applied in this study. My advice is to apply a “thematic” analysis approach for the qualitative stage. The initial phase of the mixed methods approach should involve conducting semi-structured interviews with logistics experts, Then A thematic analysis approach should apply to analyze data generated from conducting interviews with managers and experts in the logistic.

Author Response

Dear reviewer! Thank you for your encouragement and your valuable feedback which helped us to improve our paper. Following your suggestions, we hope that we were able to clarify our methodology and to refine our conclusions.

We have attached a PDF in which we go into detail about all the revisions, so based on the comments in the PDF we would ask you to review the paper. 

Reviewer 2 Report

The title needs to be changed to a more informative one.

Abstract needs to be re-written, background, aim, method, results, and conclusion should be presented in order.

The introduction needs reframing, in its current style, it is not consistent and informative. to do this task follow the stages of background: in which you should talk about the importance of the topic, literature: you should review the previous publications in this field, Research gaps: talk about the gaps in previous research and necessity of your own, aims: present your aims to fill the gaps of research, structure: mention the structure of your paper.

A flowchart is needed in the method section to clearly explain the stream of your methodology.

Then a discussion section is needed and you should compare your finding with other sources and compare the methods with each other.

The conclusion should be: A summary of the paper, highlights of your results and general conclusions from those findings, limitations of the research, and recommandations for future researchs.

References can be used to improve the paper:

https://doi.org/10.1016/j.crsust.2021.100119

https://link.springer.com/article/10.1007/s43615-021-00107-z

https://doi.org/10.1016/j.rser.2021.111968

Author Response

Dear reviewer! We highly appreciate your encouraging comments and the time which you have invested to help us improve our article. All the comments below helped us to significantly improve our paper and we now hope that we could create consistency and rigidity regarding the combination of gamified AR to raise interest in logistics professions.

We have attached a PDF in which we go into detail about all the revisions, so based on the comments in the PDF we would ask you to review the paper.

Reviewer 3 Report

Overall, I think the article is interesting, but I don't think it has enough novelty. In my opinion, both gamification and AR are not novel any more, so it is just another example of application of these techniques, well, it might be original to combine both of them, but then, the authors fail to emphasize that point and to provide more information about it. I also think the novelty could be improved by evaluating the impact of the gamification experience on the knowledge about logistics or the "attractiveness of logistical careers" (as in future work).   

The comments about the results are very interesting, I like indeed the comments about young and old participants, especially the ones about "trial and error". But, It might be very interesting to evaluate that observation quantitatively. And I also think the presentation of the quantitative results should be improved, just saying "(μ = 1.38, σ =0.6)" is clearly quite poor in order to assess the outcome. By the way, I am not so sure being above 30 should be classified as "old", ;-)  

I also miss more technical information about the development of the App, at least, it should be included: development environment and programming language. It would also be interesting to include other details about the features and details of the Apps.

Some other, more concrete comments:  

* I don't think the introduction lines from 22 to 42 are relevant in relation to the subject of the article.  

* It is not clear whether the "current shortage of skilled workers" in line 58 is either a subjective perception or an objective fact. Whatever the case should be clarified, contextualised, and explained. In line 60 and in the following sections it is also used "green logistics [careers, jobs, ...]", I supposed there is a huge variety in professional profiles in that area, so it should also be clarified and delimited.  

* Line 132, "optional" is "optionally"? or "AR maker are optional".  

* In Figure 1, I think the images should have their own figure and number. In the text, it is said that "for AR: a map in game 1 and cards in game 2", but it can not be seen in the figures o at least not clearly enough.  

* Line 257 the sentence "that the competition between younger participants is higher" is not clear. Does it mean "competitiveness"?  

* Line 370, there is an extra blank space after game 3.

Author Response

Dear Reviewer! 

Thank you very much for your positive general assessment of our paper. We would like to thank you for taking the time to read our paper so carefully and to provide useful suggestions on how to improve it. We hope we met your expectations and included your suggested revisions sufficiently in the revised vision. 

We have attached a PDF in which we go into detail about all the revisions, so based on the comments in the PDF we would ask you to review the paper.

Round 2

Reviewer 1 Report

The methodology applied in this research was not suitable

I recommend the authors didn't follow "thematic analysis ". But they didn't follow it. Hence, the research results are inaccurate.

The authors ask this question:

1) How did the players cope with game 1 "Choose the Transport ", game 2 "Transport 643 Chains" and game 3 "Green Logistics Careers"?

But in the revised document, the concept "Green logistics Careers" doesn't appear in the research.  So why they ask about "Green logistics Careers?

Author Response

Dear reviewer,

we would like to thank you for revising our paper again and for the valuable feedback. We indeed did a thematic analysis but did not have that mapped in the paper, so thank you very much for pointing out that gap. We have added the thematic analysis in the chapters on methodology and results as well as in our references. In order to improve the language style of the paper, we have had it proofread by a native speaker and corrected several phrases and typos.

We made a mistake in the description of the three games within Logistify. We called it "Green Logistics Carreers" in the paper, but the name is actually "Professions" (as you can see in Figure 1. Main menu with overview of the games). We apologise for this confusion, and we have corrected this error in the revised version.

The following table gives an overview of the above mentioned and adapted parts in the paper after the second review round:

Introduction

The summary of our method in the introduction is now described more precisely.

References

We extended our references by literature about the method of “thematic analysis”.

Methodology

We have extended the methodological description to include the description of the thematic analysis within phase 1 and phase 2 of our methodology.

Results

We explicitly address the outcomes of the conducted thematic analysis within the chapter of results.

Language and typos

In order to improve the English language style of our paper, we have had it proofread by a native speaker and changed phrasing and typos according to his recommendations.

Wording

We corrected the name of game 3 to “Professions” (instead of “Green logistics careers”)

We hope that these changes have improved the quality of our paper and that you will evaluate our work positively. 

Reviewer 3 Report

Even if I did not read this version so thoughful as the previous one, I think you have addressed and correct all issues.

Round 3

Reviewer 1 Report

Good work